# Learning better with Dale's Law: A Spectral Perspective

**Pingsheng Li***
McGill University
Mila - Quebec AI Institute
pingsheng.li@mail.mcgill.ca

**Jonathan Cornford***
McGill University
Mila - Quebec AI Institute
cornforj@mila.quebec

**Arna Ghosh**
McGill University
Mila - Quebec AI Institute
ghosharn@mila.quebec

**Blake Richards**
McGill University
Montreal Neurological Institute
Mila - Quebec AI Institute & CIFAR
blake.richards@mila.quebec

## Abstract

Most recurrent neural networks (RNNs) do not include a fundamental constraint of real neural circuits: Dale's Law, which implies that neurons must be excitatory (E) or inhibitory (I). Dale's Law is generally absent from RNNs because simply partitioning a standard network's units into E and I populations impairs learning. However, here we extend a recent feedforward bio-inspired EI network architecture, named Dale's ANNs, to recurrent networks, and demonstrate that good performance is possible while respecting Dale's Law. This begs the question: What makes some forms of EI network learn poorly and others learn well? And, why does the simple approach of incorporating Dale's Law impair learning? Historically the answer was thought to be the sign constraints on EI network parameters, and this was a motivation behind Dale's ANNs. However, here we show the spectral properties of the recurrent weight matrix at initialisation are more impactful on network performance than sign constraints. We find that simple EI partitioning results in a singular value distribution that is multimodal and dispersed, whereas standard RNNs have an unimodal, more clustered singular value distribution, as do recurrent Dale's ANNs. We also show that the spectral properties and performance of partitioned EI networks are worse for small networks with fewer I units, and we present normalised SVD entropy as a measure of spectrum pathology that correlates with performance. Overall, this work sheds light on a long-standing mystery in neuroscience-inspired AI and computational neuroscience, paving the way for greater alignment between neural networks and biology.

## 1 Introduction

Recurrent neural networks (RNNs) are a major tool in computational neuroscience research, with numerous papers that use RNNs to model the brain published every year [1, 2, 3, 4, 5]. However, when using RNNs to model real neural networks there is always the question of which biological constraints to incorporate into the model. Some biological constraints are considered law-like [6], and are thus thought to be particularly important. One example is Dale's Law, which says that neurons

---

*Equal contribution.

37th Conference on Neural Information Processing Systems (NeurIPS 2023).

release the same set of neurotransmitters at all of their axon terminals [7]. Practically, for artificial neural networks, Dale's Law implies that synaptic output weights should be all positive (excitatory, E) or all negative (inhibitory, I) for any individual neuron.[1] And, though it is common to use models that respect Dale's Law in computational neuroscience [9, 5], it is not universal practice, and it is rare outside of neuroscience. One reason for this is that incorporating Dale's Law into neural networks tends to harm their learning performance [10], which is widely known by those who have trained such networks, but rarely explicitly addressed in the scientific literature. This limits our ability to model the impressive learning of biological neural networks with RNNs that respect Dale's Law. It also impairs studying the unique computations played by inhibitory cells using RNN models [11, 12, 13], and restricts comparisons between RNN models and real neural data [14, 15].

Typically, Dale's Law is incorporated into RNNs using the most obvious solution, by constraining entire columns of the synaptic weight matrices to be of the same sign, thereby partitioning the hidden units into E and I populations (here we refer to these networks as 'Column Excitation-Inhibition' or 'ColEI', see Figure 1 for an illustration) [9, 5, 16, 17, 18, 19]. While simple and intuitive, as mentioned, this approach impairs learning performance [10]. One hypothesis for why this is, is that EI sign constraints impair network performance by limiting the network's solution space [20, 10]. Indeed, a recent feedforward EI ANN architecture (Dale's ANNs, or DANNs), inspired by fast inhibitory circuits in the brain, and which was designed to overcome sign constraints, was found to learn just as well as standard ANNs [10, 21]. In this work, we show that recurrent versions of DANNs can also learn as well as standard RNNs. But, we also want to determine whether the sign constraints are really the primary cause of impaired learning in ColEI networks. Are there other reasons that ColEI networks learn worse than standard RNNs and DANNs?

Here we show that the *spectral properties* of EI networks that are most important for their performance. We find that the singular value spectrum of ColEI networks as in [9] is multimodal, dispersed, and includes a notable mode of large singular values. In contrast, DANNs and standard RNNs exhibit unimodal spectra as expected from random matrix theory [22]. Next, we transplant spectra between networks and find the spectrum is more impactful than sign constraints on performance - though both contribute. Furthermore, we find that small ColEI networks with biologically realistic E to I units ratio learn less effectively, and this is reflected in their spectral properties. In contrast, the spectrum and learning of DANNs is robust to both the ratio of E and I units, and network size: even small networks with only 1 inhibitory neuron have good spectral properties and learn well. Finally, we show that normalised SVD entropy, a measure of unimodal singular value clustering, is predictive of EI RNN learning performance.

Altogether, our results provide evidence that the reason for the impaired performance of traditional ColEI RNN networks is predominantly their spectral properties, and less so their sign constraints. Therefore, the improved performance of DANNs should be understood as primarily resulting from them having spectral properties that are similar to standard RNNs, in addition to their being robust to sign constraints. This is a critical insight for designing EI networks that are architecturally different from DANNs, which may be desirable when modelling specific subtypes of interneurons [23]. Finally, more broadly we present an answer to the long-standing mystery of poor performance of RNNs that respect Dale's Law, thereby paving the way for their future development and use in neuroscience-inspired AI.

## 2 Methods

### 2.1 Model Definitions

In this section we first define "standard" RNNs, then the two different kinds of RNNs that obey Dale's Law, namely ColEI and DANNs. To obey Dale's Law, ColEI and DANN networks were first initialised in line with Dale's law (see initialisation details below), and projected gradient descent was used to maintain parameter signs during training. As such, after a candidate gradient descent update, parameters that had changed sign were set to 0, corresponding to a euclidean projection onto to the allowed solution space.

The reader can find the code for all our experiments available here.

---

[1] See [8] for a rare exception of excitatory and inhibitory neurotransmitter co-release.

**Standard Recurrent Neural Networks (RNN)**

For a multilayer network, we define the hidden state activations of a "standard" RNN layer $\ell$ at time $t$, $h_t^\ell$, via:

$$h_t^\ell = \sigma(W_{in}^\ell h_t^{\ell-1} + W_{rec}^\ell h_{t-1}^\ell + b^\ell) \tag{1}$$

where $\sigma(\cdot) = ReLU(\cdot)$ is the activation function, $h_t^{\ell-1}$ is the activation vector for the lower layer, $h_{t-1}^\ell$ is the hidden state activations at the previous time-step, $W_{in}^\ell$ and $W_{rec}^\ell$ are the input and recurrent synaptic weight matrices, and $b^\ell$ is the bias. Note $h_{t-1}^\ell$ is equivalent to the input, $x_t$, for the first layer, i.e. $h_t^0 = x_t$. The output, $y_t$, is a function of the final layer's ($L$) hidden state:

$$y_t = g(W_{out} h_t^L + b_{out}) \tag{2}$$

where $W_{out}$ is the output weight matrix, $b_{out}$ is the output bias, and $g(\cdot)$ is an optional output non-linearity. We set $g(\cdot)$ depending on the task: softmax function for classification and language modelling tasks, and identity for the adding problem.

**Column Excitation-Inhibition (ColEI)**

The ColEI model discussed in this work has the same formulation as a standard RNN (equations 1 and 2) except that the weight matrices are sign constrained. Specifically: (1) the input weights of the first recurrent layer $W_{in}^{(1)}$ are all positive; (2) all other weight matrices are constrained to have columns whose entries are the same sign, either all positive or all negative. In combination with ReLU activations, this constraint makes positive columns correspond to the output weights of presynaptic excitatory neurons, and negative columns inhibitory neurons. An illustration is provided in Figure 1 A.

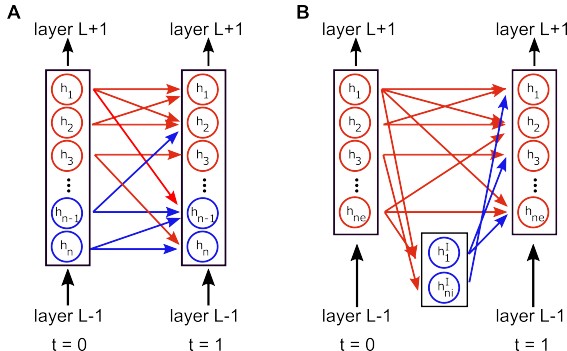

Figure 1: Illustration of ColEI and DANN weight matrix partitions with excitatory (red) and inhibitory (blue) populations. (A) The ColEI partition, which separates the neurons within a layer into E and I populations. (B) The DANN partition with a population of I neurons placed between E layers.

**Dale's Artificial Neural Networks (DANN)**

DANNs are an EI architecture first introduced in [10] for feedforward networks that were inspired by fast feedforward inhibition the brain [24]. Here we extend DANNs to the recurrent setting. A DANN layer reparameterises the weight matrices of standard ANNs using three separate non-negative matrices. Specifically, for a given weight matrix $W$, we have:

$$W = W^{EE} - W^{EI}W^{IE} \tag{3}$$

where $W^{EE}$, $W^{IE}$, and $W^{EI}$ denote E-to-E, E-to-I, and I-to-E projections respectively (see Figure 1 B for an illustration). Note that such a parameterisation allows the sign of each element of $W$ to be unconstrained, although the signs of individual weight matrices, i.e $W^{EE}$, $W^{IE}$, and $W^{EI}$, are constrained by clipping illegal weights to 0 after each update.

## 2.2 Initialisation details

**Standard RNNs:** The weights of standard RNNs were sampled from a uniform distribution $U[-\frac{1}{\sqrt{n}}, \frac{1}{\sqrt{n}}]$, where $n$ is the number of hidden units (i.e. the PyTorch default initialisation), unless otherwise stated. Biases were initialised as 0.

**ColEI:** The parameters of ColEI networks were initialised following [10]. The weights in each column were sampled from either the exponential distribution corresponding to inhibitory (negative) or excitatory (positive) weights. The distribution means were set (i) to ensure that the sum of excitatory and inhibitory inputs to each neuron was balanced in expectation (i.e. $\sum_{j \in exc} |\mu_j| = \sum_{j \in inh} |\mu_j|$ for inputs indexed by $j$) and, (ii) following [25, 26], so that activation variance did not scale with depth (see Appendix 6.6 for details). Next recurrent weights were adjusted to constrain their spectral radius, in line with [9], by setting $W^{\ell}_{rec} \leftarrow (\rho/\rho_0) W^{\ell}_{rec}$, where $\rho_0$ is the spectral radius of $W^{\ell}_{rec}$ before the adjustment, and $\rho = 1.5$ unless stated. Biases were initialised as 0, except for the first layer, where they were set to centre the activations.

**DANNs:** Unless explicitly mentioned, all entries of $W^{EE}$ matrices were independently sampled from an exponential distribution of variance $\frac{n}{3n(n-1)}$, where $n$ is the number of hidden units. Next, to balance inhibitory and excitatory inputs, each row of $W^{IE}$ was initialised as the mean row of $W^{EE}$ (therefore each row of $W^{IE}$ is the same at initialisation), and rows of $W^{EI}$ were sampled from an exponential distribution and then squashed so that the sum of each row was 1. As such, the variance of $W$ is the same as Pytorch's default initialisation for RNNs. Biases were initialised as 0.

## 2.3 Experiments

We tested the three different types of networks (standard RNN, ColEI, and DANN) on three classical tasks for RNNs: the adding problem [27], sequential MNIST classification [28], and language modelling using the Penn Tree Bank [29] All experiments were run with PyTorch version 1.5.0 on a RTX 8000 GPU cluster. Unless otherwise stated, all presented results were averaged over 5 seeds and shaded error bars denote standard deviation. Performance metrics are plotted after the first 100 updates.

**Adding problem:** The adding problem serves as a test to evaluate the ability of RNNs to learn long-term dependencies [27]. The objective is to compute the sum of two numbers selected from a sequence of random numbers $\in [0, 1]$ (of length 20 in our experiments). At each time step the input to the network is a 2-dimensional vector, comprising an element from the sequence of random numbers and a boolean, which is one only for the two numbers that should be added and zero otherwise.

**Sequential MNIST:** The sequential MNIST problem is a common test for the ability of RNNs to engage in classification. In this task, the network must classify MNIST digits (i.e. handwritten digits of numbers) when the rows of the image are presented sequentially to the network from top to bottom (resulting in 28 timesteps).

**Penn Tree Bank:** The Penn Tree Bank task is a common baseline for natural language processing that is more challenging than either the adding problem or sequential MNIST. In this task, input words from sentences taken from a newspaper corpus are embedded as 300-dimensional vectors using GloVe [30]. At each time step, the network must predict the next word.

**Natural object recognition task:** Following [31], we designed the following naturalistic object recognition task. At the first time step, the network's input is a down-sampled, flattened achromatic image of one of 64 natural objects from 8 categories (e.g. fruits, faces, etc.) with a random natural-scene background. The network receives no further inputs and must retain information about the image over 5 time steps to output the classification result at the last time step. Please see [31] and Appendix Fig. 8 for more details.

### 2.3.1 Spectrum Transplant

Singular value decomposition (SVD) decomposes a matrix into matrices of singular values and singular vectors: $W = U\Sigma V^T$, where $U$ and $V$ are orthonormal matrices whose columns are singular vectors, and $\Sigma$ is a matrix of singular values. Therefore, network weight matrices can be decomposed as: $W_{RNN} = U_{RNN}\Sigma_{RNN}V^T_{RNN}$, $W_{ColEI} = U_{ColEI}\Sigma_{ColEI}V^T_{ColEI}$. To disentangle the impact of singular values and singular vectors on performance we constructed networks initialised with weight

matrices $W_1 = U_{RNN} \Sigma_{ColEI} V_{RNN}^T$ and $W_2 = U_{ColEI} \Sigma_{RNN} V_{ColEI}^T$, such that a standard RNN was given the spectral properties of a ColEI network, and vice-versa.

### 2.3.2 Normalised SVD Entropy (NSE)

To quantify the clustering strength of the singular values of a $N \times N$ recurrent weight matrix we used normalised SVD entropy (NSE), also denoted as $H$ [32]:[2]

$$H = -\frac{1}{log(N)} \sum_i^N \bar{\sigma}_i log(\bar{\sigma}_i), \quad \text{where} \quad \bar{\sigma}_i = \frac{\sigma_i}{\sum_j^N \sigma_j} \qquad (4)$$

where $\sigma_i$ are the singular values. The normalisation term $log(N)$ ensures $H \in [0, 1]$ regardless of network size, where $H = 1$ if all singular values are the same (clustered) and $H \approx 0$ implies the distribution of singular values is extremely dispersed. For example, in orthogonal RNNs $H = 1$, since all singular values of the recurrent weights are 1 [33].

## 3 Results

### 3.1 Network performance on benchmark tasks

We first assessed the performance of standard, ColEI, and DANN RNNs on four benchmark tasks for RNNs: the adding problem, row-wise sequential MNIST, the Penn Tree Bank dataset, and a naturalistic object recognition task (Section 2.3). We found that ColEI networks did not learn as well as DANNs and standard RNNs. Their final performance was worse across all four tasks, and more variable across random seeds (Figure 2 A-D). Furthermore, we observed that gradient clipping benefitted ColEI networks substantially more than the other networks (Appendix Figure 10).

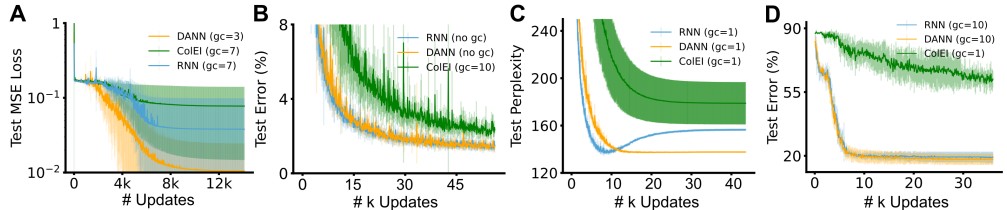

Figure 2: Performance of networks on benchmark datasets. (A) Adding problem (1 layer of 10 neurons). (B) Sequential MNIST (3 layers of 100 neurons). (C) Penn Tree Bank (3 layers of 500 neurons). (D) Naturalistic Object Recognition (1 layer of 1000 neurons). $gc$ denotes gradient clipping.

A confound that could explain the discrepancy in performance between ColEI networks and DANNs & standard RNNs is the distinct strategies used for their initialisation (Section 2.2). Specifically, ColEI recurrent weights are initialised such that the greatest norm of eigenvalues, i.e. the spectral radius $\rho$, is 1.5 [9]. In contrast, the initialisation of DANNs & standard RNNs results in $\rho \simeq 1/\sqrt{3}$ (PyTorch default RNN initialisation). Therefore, we assessed model performance across a range of initial values of $\rho$ (Appendix 9). Again, we found that ColEI networks learned poorly, and there was no value of $\rho$ for which ColEI networks matched the performance of the best standard RNNs and DANNs. However, while standard RNNs and DANNs had a similar relationship between $\rho$ and performance, ColEI networks performed better than regular RNNs and DANNs at larger values of $\rho$, and worse for smaller values (Appendix Figure 9). Additionally, we also ran a control experiment to verify that the inhibitory parameter scaling of DANNs did not improve ColEI network performance (Appendix Figure 11).

### 3.2 Spectral properties of networks

The different relationship between performance and $\rho$ of ColEI networks vs standard & DANN RNNs (Figure 2 D) is indicative of different spectral properties, such as the eigenvalue distribution of the recurrent weights. For example, eigenvalues greater (or less) than one indicate activity directions that

---

[2]Note this quantity is not the entropy of the probability distribution of singular values.

lead to exploding (or vanishing) activations and gradients, thereby impairing learning. Inspecting the eigenvalues of the initial weight matrices in the complex plane (Figure 3 A-C) shows that the eigenvalues of ColEI networks follow a non-uniform distribution that is clustered around the origin, as has been previously reported [34, 35]. In contrast, the eigenvalue distributions of DANNs & standard RNNs converge to a uniform circle in the complex plane.[3] We note that these spectral properties provide an informal explanation for the better learning of ColEI networks at larger values of $\rho$ (Figure 2 D), as there will still be a majority of smaller eigenvalues in addition to the minority of large eigenvalues.

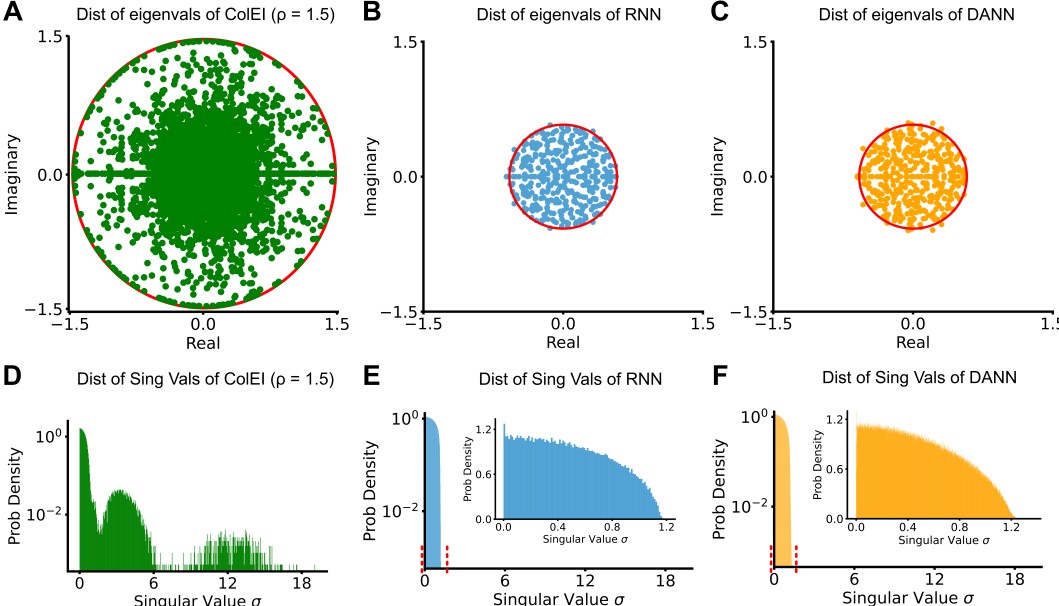

Figure 3: Visualisation of eigenvalues and singular values for ColEI, standard, and DANN recurrent weight matrices. (A-C): Eigenvalues plotted in the complex plane for 100 different initialisations. Red circle in A has a radius of 1.5; for B,C radius $= 1/\sqrt{3}$. (D-F): Histograms of singular values for 100 different initialisations. Insets in E,F show blow-ups of the red lines in main plot.

However, there are limitations to using eigenvalues for understanding learning dynamics. Importantly, the eigenvectors of non-symmetric matrices are always non-orthogonal, and eigendecomposition may fail to capture certain features of matrices with highly non-orthogonal columns – such as ColEI networks [36]. In contrast, singular vectors are always orthogonal. Thus, for ColEI networks we reasoned that although the maximum eigenvalue is constrained to be 1.5, the maximum singular value may be large and unconstrained. Indeed, while the maximum singular value for DANNs & standard RNNs was approximately twice the maximum radius of their eigenvalues, the ColEI singluar value spectrum had a notable mode of large singular values (Figure 3 D-F).[4] Also, the distribution of singular values for ColEI networks was multimodal (Figure 3 D-F), whereas the distribution of standard RNNs and DANNs converged to a quarter circle, in agreement with the Marchenko-Pastur Quarter Circle Law[5]. These results tell us that even when the eigenvalue radius is constrained for ColEI networks, there will still be directions in activity space corresponding to the large singular values along which the activations and gradients will be heavily deformed and stretched, leading to poor learning.

---

[3]This is a corollary of Girko's circle law, which states that, for large N, the eigenvalues of an $N \times N$ random matrix are distributed uniformly within the complex unit circle, if the elements are independently sampled from a zero-mean distribution with variance $\frac{1}{N}$. Pytorch default initialisation ($U[-\frac{1}{\sqrt{N}}, \frac{1}{\sqrt{N}}]$) has a variance of $\frac{1}{3N}$, which scales down the radius of the circle by $\frac{1}{\sqrt{3}}$.

[4]This phenomenon is not unique to exponential distribution, which we used in the initialisation of ColEI RNNs. We also observed similar spectral properties for gamma and uniform distributions - see Appendix 14.

[5]This result is for a $N \times N$ matrix with i.i.d. elements drawn from a distribution with zero mean and variance [22], and is agnostic to the exact distribution from which the elements are drawn.

## 3.3 Spectral properties matter more than sign constraints for learning

A common hypothesis is that the poor performance of EI networks is due to the parameter sign constraints applied during learning, and addressing this was a motivation behind the architectural design of DANNs [10, 20]. Given the different spectral properties of ColEI and standard RNNs that we observed, we wondered: are sign constraints the main reason for impaired learning in ColEI networks, or are the spectral differences from standard RNNs the most critical factor? To investigate this, we first explored the impact of sign constraints: we trained ColEI networks without sign constraint during learning, and we trained standard RNNs with parameter signs frozen after initialisation (Figure 4 A, sequential MNIST). As expected, for both networks we found that sign constraints impaired learning ($0.7\% \pm 0.3\%$ for RNNs, $1\% \pm 1\%$ for ColEI), but they did not account for the full difference in performance between standard RNNs and ColEIs ($2.3\% \pm 0.6\%$).

Next, we designed spectrum "transplant" experiments, to investigate the impact of singular value spectra on learning (Section 2.3.1). We constructed hybrid networks with a combination of spectral properties at initialisation and sign constraints (see Section 2.3.1), specifically: 1) RNN or ColEI spectrum, 2) RNN or ColEI singular vectors, 3) presence or absence of sign constraints during learning. In total, with these three factors we had eight different network configurations ($2^3 = 8$ networks). These experiments revealed that the singular value spectra contribute more than sign constraints and singular vectors to the performance gap between ColEI and RNNs (Figure 4 B,C $1.5\% \pm 0.3\%$ for RNNs, $2\% \pm 1\%$ for ColEI). In addition to sequential MNIST, we also ran the same set of experiments on the Naturalistic Object Recognition task, and observed very similar results (Appendix Table 2). Finally, we investigated if the empirical EI balance at initialisation (i.e. the mean of the sum of incoming weights to each neuron) was altered in these experiments. We found it was not: for a ColEI network with 10% inhibitory neurons and 1000 hidden units, the mean balance across units was -0.0006 with std. deviation 1.12, and for ColEI with the RNN spectrum it was 0.0696 with std. deviation 0.88 (t test, p=0.12). Thus, sign constraints are only one component, and the singular value spectrum at initialisation has a stronger impact on learning performance.

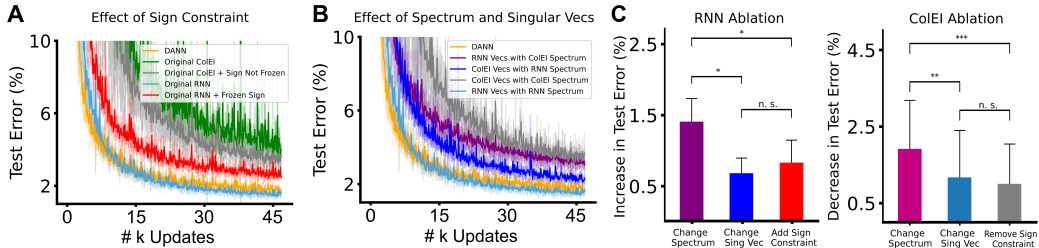

Figure 4: Impact of spectrum, singular vectors and sign constraints on learning (A) Impact of sign constraints on standard and ColEI RNNs (note that DANNs are always sign constrained). (B) Impact of transplanting singular vectors and values between ColEI and RNN networks. (C) Summary of final test error changes induced by spectrum, singular vectors, and sign constraints. RNNs 5 seeds ($p < 0.05$); ColEI 30 seeds due to higher variance ($p < 0.01$). Mann–Whitney U test

## 3.4 Network performance and spectrum with respect to changes in EI ratio and network size

Having established that the spectrum of singular values is important for ColEI learning, we investigated how architectural hyperparameter choices, such as the EI ratio and number of hidden units, change the spectral properties and learning performance. Here, to quantify changes in the singular value distribution, we leveraged normalised SVD entropy (NSE) as a measure of spectrum "pathology" (see Section 2.3.2). Notably, NSE captures the degree of unimodal clustering in the singular value spectrum, and we hypothesise that a unimodal spectrum of clustered singular values is desirable - as is the case for orthogonal RNNs [33, 28].

We found that reducing the percentage of inhibitory neurons (thereby increasing the EI ratio) decreased the NSE and impaired learning performance (Figure 5). Also, despite constraining $\rho = 1.5$ in each case, decreasing the number of inhibitory neurons increased the largest singualar value, $\sigma_{max}$ (Figure 5 D-F). In contrast, we found that the spectra and performance of DANNs were robust to changes in the EI ratio, and they learned well even with 1 inhibitory neuron (NSE remained $\sim 0.95$ for both DANNs and standard RNNs, Figure 5 A-C, Appendix Figure 15, Appendix Table 3). This

result further supports the idea that the spectral properties of EI networks predict their learning performance, and shows that the E-to-I ratio impacts the spectral properties of ColEI networks.

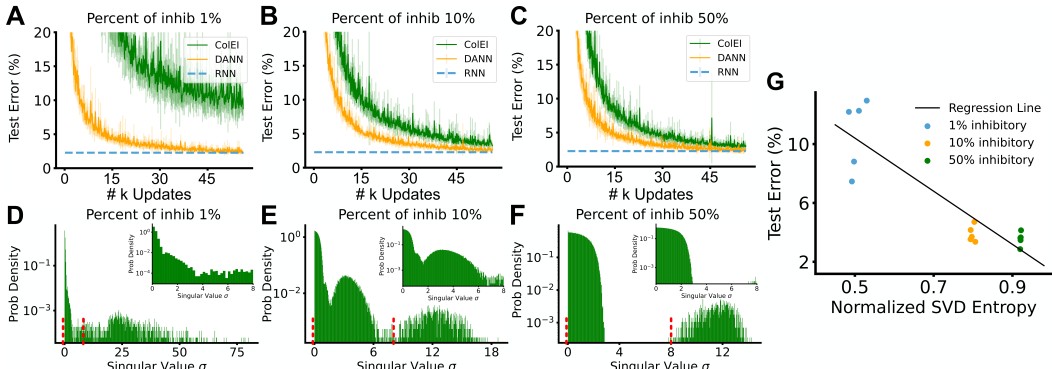

Figure 5: Performance and spectral properties of DANN and ColEI RNNs with different EI ratios. (A-C) Performance on sequential MNIST. Dashed line indicates the final error of standard RNNs of same size. (D-F) Distribution of singular values for ColEI networks y-axis log scaled, inset plots are x-axis $[0, 8]$. See equivalent for DANNs in Figure 3 F and Appendix Figure 15 (G) ColEI performance v.s. initial normalised SVD entropy of the recurrent weights across different EI ratios. (1-layer RNN, 100 units, $\rho_{ColEI} = 1.5$, $\rho_{DANN} = \frac{1}{\sqrt{3}}$

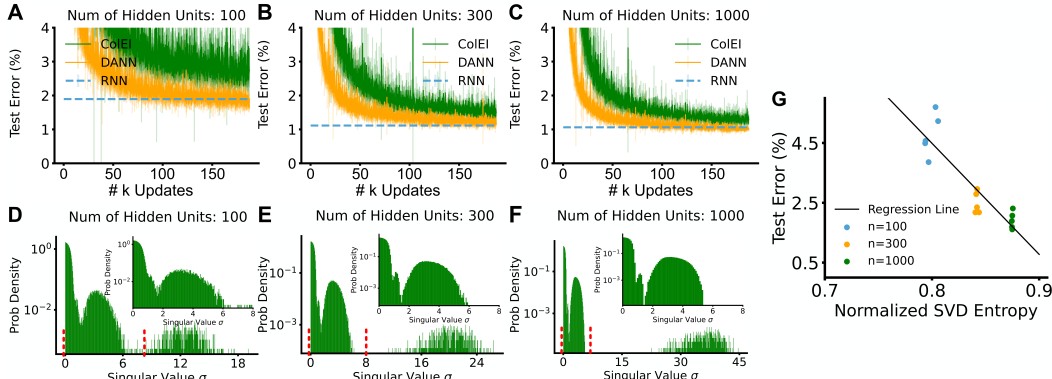

Figure 6: Performance and spectral properties of DANN and ColEI RNNs with different network sizes (column-wise). (A-C) Performance on sequential MNIST. Dashed line indicates the final error of RNNs of same size. (D-F) Distribution of singular values for ColEI networks y-axis log scaled, inset plots are x-axis $[0, 8]$. See equivalent for DANNs in Appendix Figure 15 (G) ColEI performance v.s. initial normalised SVD entropy of the recurrent weights across different network sizes. (1-layer RNN, 10% inhibitory units, $\rho_{ColEI} = 1.5$, $\rho_{DANN} = \frac{1}{\sqrt{3}}$

Next, we investigated how the number of hidden units affects the spectral properties and learning performance of EI networks. As shown in Figure 6 A-F, we found that the bulk of singular values became more clustered for larger ColEI networks ($\in [0, 8]$ Figure 6 D-F), and the NSE increased despite an increasing $\sigma_{max}$. Correspondingly, these changes in NSE were anti-correlated with test error (Figure 6 G), similar to the performance trends for EI ratio. In contrast, DANN spectra were similar across network sizes (Appendix Figure 15). Taken together with the EI ratio experiments, these results suggest that for ColEI networks, the best performance will be obtained for larger networks with balanced EI ratios, because these networks have spectral properties more similar to standard RNNs. However, it is interesting that large ColEI networks learn well despite a large $\sigma_{max}$, which indicates that the overall effect of all singular values, rather than $\sigma_{max}$, is what matters most for learning. We futher discuss this result in the discussion, but put another way, in larger networks even if some directions have large singular values, there are many more directions with small singular

values. As a result, activities will likely be rotated away from the directions with large singular values, ameliorating the exploding gradients problem. Therefore, for a metric to be predictive of the network performance it should capture the overall effect of the entire spectrum, which we do with NSE.

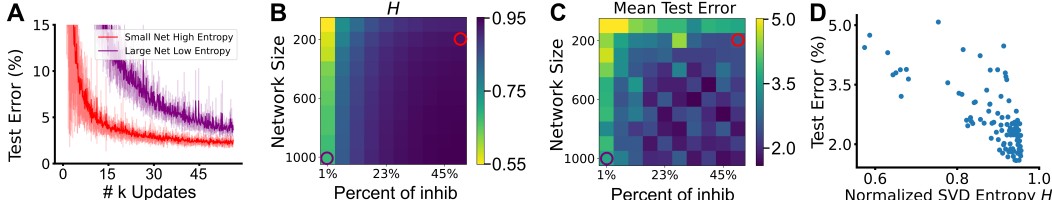

Figure 7: Normalised SVD entropy predicts ColEI network performance. (A) Performance on sequential MNIST for networks indicated by circles in B&C. (B) Normalised SVD entropy for networks of different sizes and EI ratios. (C) Mean test error for networks of different sizes and EI ratios. Top left pixel value is clipped at 5 for better visualisation. (D) Final mean test error vs Normalised SVD entropy for all ColEI networks. 1 outlier network (corresponding to the top left pixel in C, B) is removed for better visualisation.

The above experiments suggest that the spectral properties of ColEI networks at initialisation, summarised by NSE, can be used to estimate their performance across a range of architectural hyperparameters. As the NSE at initialisation is dependent on both the network size and EI ratio, networks can have a high NSE from either being large, or having a balanced EI ratio. Therefore, we hypothesised that small networks with balanced E & I units will outperform larger networks with a skewed EI ratio that results in a smaller NSE at initialisation. To test this prediction, we generated networks of different EI ratios and numbers of hidden units and calculated their associated NSE at initialisation Figure 7 B. Again, we found that NSE anti-correlates with final test error (Figure 7 D), and small networks with high NSE performed better than large networks with lower NSE (Figure 7 A). Similar trends are observed for the natural object recognition and language modelling tasks (Appendix Figure 12 & Figure 13 respectively). This supports the hypothesis that the NSE at initialisation is of primary importance for determining learning performance. Moreover, it allows us to characterise the link between spectrum pathology and performance and explain why different hyperparameter choices may affect learning in EI RNNs.

## 4 Discussion

In this work, we have provided evidence that both sign constraints and the spectral properties of the weights at initialisation can cause poor learning in EI RNNs. As well, we compared the relative contributions of sign constraints and spectral properties for EI network performance, and found that the distribution of singular values are more important for learning. We found that the simplest method of incorporating Dale's Law into RNNs (by constraining columns of the weight matrix, ColEI, following [9]) results in networks with multi-modal spectra that are dispersed, especially for small networks with large ratios of E to I units. In contrast, we found that recurrent versions of the DANN EI architecture [10] have similar spectra to standard RNNs, and learn just as well. We also presented NSE as a measurement of spectrum pathology that can be used as a diagnostic tool to predict performance before training. The higher the NSE at initialisation, the more likely an EI RNN is to train well. This work therefore explains why DANN RNNs perform as well as standard RNNs, but not ColEI RNNs. Overall, by highlighting the importance of EI network spectral properties at initialisation we provide a map for designing networks that respect Dale's law and learn well.

In addition to their poor learning, ColEI networks also showed more variance during training and over different initialisations when compared to the other networks. This is likely due to them having fewer activity modes with appropriate singular values for learning computations over time (the motivation behind SVD entropy) and therefore small differences at initialisation can strongly impact learning. Similarly, we also found that without gradient clipping ColEI networks are highly unstable, again in line with a pathological distribution of singular values at initialisation [37]. However, other open theoretical questions arise from the empirical results presented above. First, what is the origin of the unusual multi-modal spectra of ColEI networks? And second, why do large ColEI networks learn well despite a large maximum singular value, $\sigma_{max}$?

For a simple intuitive explanation of ColEI spectrum pathology, let us first consider the difference between the recurrent weight matrix of ColEI and standard RNNs. For both the standard & DANN RNNs, this a zero-mean random matrix. However, the ColEI recurrent weight matrix can be written as the sum of a rank-one nilpotent matrix (of column means) and a zero-mean random matrix. While technically the singular and eigen value distributions of two matrices can't simply be added when two matrices are added, this already provides some intuition as to the origin of the unusual spectrum: it likely stems from the nilpotent matrix of column means. Here we note that the eigenvalues of any nilpotent matrix are 0, and hence we expect they do not significantly change the eigenvalue radius. If so, constraining the spectral radius of the recurrent matrix will not constrain the nilpotent matrix's contribution towards the singular value spectrum (see the Appendix 6.5 for a more detailed discussion).

Regarding the second open question, we found that large ColEI networks learned well, despite an increasing $\sigma_{max}$. Intuitively this should harm learning, as some activity directions will result in exploding gradients [37]. However, we can understand this by writing the operation performed by a linear network in terms of the SVD of its recurrent weight matrix:

$$h_{t+2}^\ell = W_{rec}^\ell W_{rec}^\ell h_t^\ell = U\Sigma V^T U \Sigma V^T h_t^\ell \tag{5}$$

It is clear that between each scaling operation $\Sigma$, there are two rotation steps $V^T, U$. These rotation steps could allow the network to avoid pathological growth by rotating the activity vectors aligned to directions with large singular values, such that they instead align with directions with small singular values. Furthermore, as the network width $n$ increases, the number of rotations where this effect occurs increases, as there exist more directions with smaller singular values, as highlighted by NSE.

There are several avenues for future work that build on the results we present here. First, there may be other ways to design RNNs that respect Dale's Law while learning well. Our results suggest that a guiding principle for investigations of this sort should be the consideration of the spectral properties of the weight matrices. And further, we note that this insight likely applies broadly to all EI network architectures, such as convolutional, feedforward, and potentially spiking models, not just RNNs. Similarly, the models we present here could be made more biologically faithful, for example inhibitory to inhibitory connections could be added to DANNs, and cell-type specific time constants included in ColEI networks. Furthermore, the two EI models have different neurobiological interpretations, as the populations of inhibitory neurons that they naturally model are different. ColEI RNNs are a more general model of inhibitory cells (eg. SOM+, CCK+ etc.), whereas DANNs specifically model fast PV+ interneurons. Therefore a promising future line of work could be to hybridise the two the models together. As our central motivation was to understand why EI networks can fail to learn well, we didn't explore these directions in the experiments we present here. However, our results indicate that the network's spectral properties should be carefully considered whenever such architectural choices are made for modelling purposes.

Additionally, from a biological-hypothesis generating perspective, appropriately constraining the spectral properties of RNNs can help ameliorate problems of exploding/vanishing gradients [36], and so, an interesting question for neuroscience research into biologically plausible gradient estimation [38] is whether real brains, which largely obey Dale's Law, have spectral properties that are well-suited to gradient descent or not. Likewise, there are interesting questions regarding the balance between excitation and inhibition in RNNs (a major topic in neuroscience) that flow from the observation that such balance can impact the spectral properties of networks [11].

In summary, our work builds on tools from deep learning theory to provide guidance on how to construct RNNs that incorporate Dale's Law. We presented evidence that the reason for poor learning of ColEI networks and good learning of DANNs is, in large part, their respective spectral properties. Therefore, we hope our work serves as the basis and inspiration for an alternative initialization for ColEI networks that improves their learning. More generally, by providing an explanation for EI network learning differences, we have opened up new directions of research for RNNs in computational neuroscience.

# 5   Acknowledgements

This work was supported by a NSERC (Discovery Grant: RGPIN-2020-05105; Discovery Accelerator Supplement: RGPAS-2020-00031; Arthur B. McDonald Fellowship: 566355-2022) and CIFAR (Canada AI Chair; Learning in Machine and Brains Fellowship). This research was enabled in part by support provided by (Calcul Québec) (https://www.calculquebec.ca/en/) and the Digital Research Alliance of Canada (https://alliancecan.ca/en). The authors acknowledge the material support of NVIDIA in the form of computational resources. JC: IVADO Postdoctoral Fellowship and the Canada First Research Excellence Fund. AG: Vanier Canada Graduate scholarship. PL wishes to convey his deepest gratitude to Ann Z.X. Huang for her support.

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
