## 6.2 Experimental Details

### 6.2.1 Adding Problem

The batch size is 32. The number of time step is 20. The loss function is mean-squared-error loss. The learning rates and the magnitude of gradient clipping are tuned by grid search over {0.5, 0.1, 0.05} and {1, 3, 5, 7, None}. The learning rate decays by a factor of 0.99 after every 20 updates. All RNNs have one recurrent layer of 10 hidden units and one linear readout layer. The numbers of samples used for training, validation, and testing are: 8000, 1000, and 1000 respectively. Hyper-parameters that give the best performance on the validation set are used for testing.

### 6.2.2 Sequential MNIST Classification

The batch size is 32. The number of time step is 28. The loss function is cross-entropy loss. The learning rates and the magnitude of gradient clipping are tuned by grid search over {1e-1, 1e-2, 1e-3, 1e-4, 1e-5} and {1, 5, 10, None}. For the first experiment, all RNNs have 3 recurrent layers and one linear layer of 100 hidden units. For the experiment with different proportions of inhibitory neurons, all RNNs have one recurrent layers and one linear layer of 100 hidden units. For the experiment with different network widths, all RNNs have one recurrent layer of 100, 300, or 1000 hidden units and one linear readout layer. The numbers of samples used for training, validation, and testing are: 50000, 10000, and 10000 respectively. Hyper-parameters that give the best performance on the validation set are used for testing.

### 6.2.3 Language Modelling using Penn Tree Bank

The batch size is 64. The number of time step is 50. The loss to be minimized is perplexity: i.e. exponentiated cross-entropy loss. The learning rate and the magnitude of gradient clipping are tuned by grid search over {3, ..., 0.03} and {1, 5, 10, 50, None}, and the learning rate decays by a factor of 0.99 after every 50 updates. All RNNs have 3 recurrent layers and 3 linear layers of 500 hidden units. The numbers of words used for training, validation, and testing are: 929536, 73746, 82416 respectively. Spare words are discarded. Hyper-parameters that give the best performance on the validation set are used for testing.

### 6.2.4 Naturalistic Object Recognition

The batch size is 16. The number of time step is 5. The loss function is cross-entropy loss. The grid search method is employed to fine-tune the learning rates and gradient clipping magnitude, considering values from the sets {1e-1, 1e-2, 1e-3, 1e-4, 1e-5} and {1, 5, 10, None}, respectively. all RNNs have 1 recurrent layer followed by a linear readout layer. Images are are down-sampled by a factor of 3 via bicubic interpolation. Initially, the network is presented with a flattened image that represents one of 64 natural items from 8 distinct categories, such as Fruits or Faces, superimposed on a random natural background. Over the subsequent five timesteps, with no additional inputs, the network must maintain the image's information to ultimately classify it during the final step. This task is challenging due to two requirements: the need to preserve image information over extended periods without new inputs and the challenge of correctly categorizing varying objects, like raspberries and watermelons, under a shared label like Fruits. The numbers of samples used for training, validation, and testing are: 512, 64, and 64 respectively. Hyper-parameters that give the best performance on the validation set are used for reporting performance.

## 6.3 Optimisation

All models were trained with Back Propagation Through Time (BPTT), without truncation [39]. In all experiments gradient clipping was used, i.e. gradients were rescaled such that their norm did not

exceed a threshold, $gn_{max}$ ($\|\nabla L\| \leq gn_{max}$), where $L$ is the dataset-specific loss function. The value of $gn_{max}$ was obtained by hyperparameter search.

DANN and ColEI models were trained with projected stochastic gradient descent (SGD) in order to maintain sign constraints, i.e. SGD followed by a rectification if the updated weight changed sign. Following [10], for DANNs, in order to balance the impact of E and I weight updates, the learning rates for $W^{EI}$ and $W^{IE}$ matrices were scaled by $\frac{1}{n_{in}}$ and $\frac{1}{\sqrt{n_{out}}}$, where $n_{in}$ and $n_{out}$ denote the dimension of input and output respectively.

## 6.4 Supplementary results

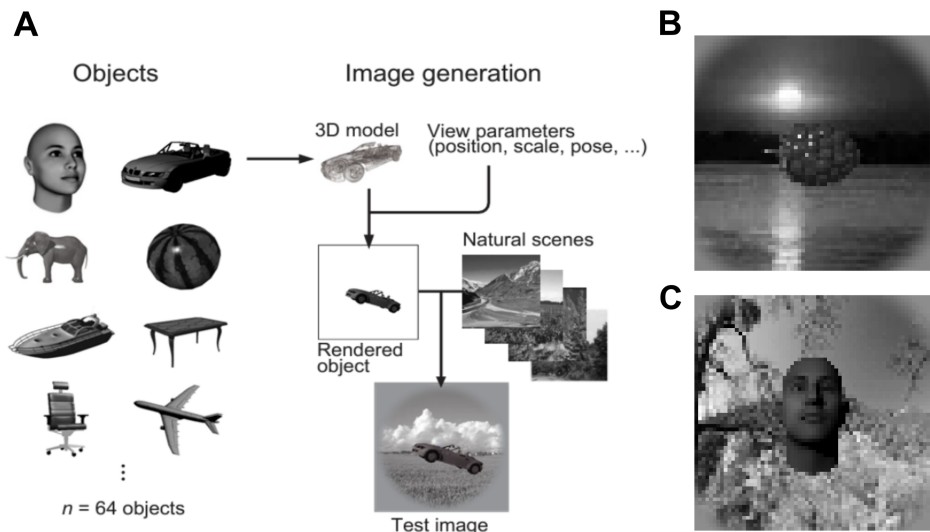

Figure 8: Naturalistic object recognition task description and results. Images were presented to the networks only at the first time-step, and the networks processed this for five time-steps before outputting the category. (A) Pipeline for generating images. Reproduced from [31] (B) Examples of images (raspberry and face) used in our experiment.

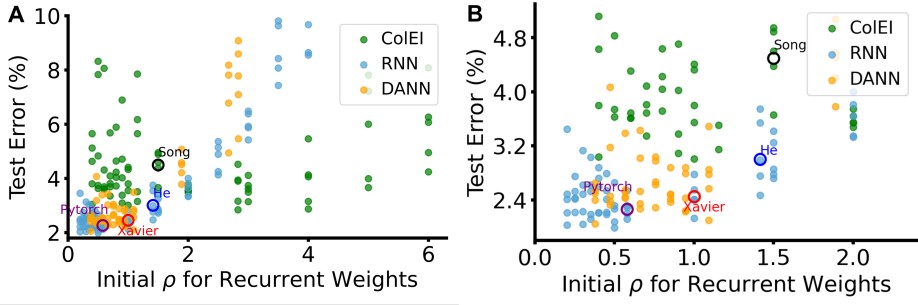

Figure 9: (A) Performance on sequential MNIST vs weight spectral radii $\rho$ at initialisation (1 layer of 100 neurons, $\rho \in [0.4, 6]$ for ColEI, DANN, $\rho \in [0.2, 6]$ for RNN). gc denotes gradient clipping. (B) Zoomed-in version of (A).

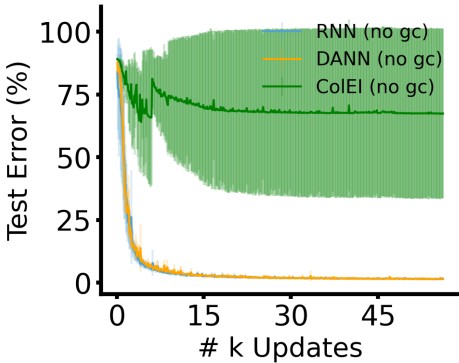

Figure 10: Network performance on SeqMNIST without gradient clipping.

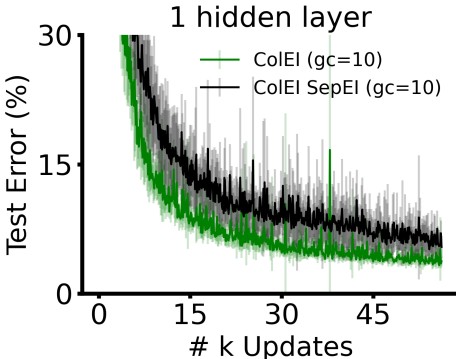

Figure 11: Performance on SeqMNIST for ColEI networks with (black) and without (green) scaled learning rates for inhibitory parameters as in DANN networks.

| Models | ColEI Spectrum | ColEI Singular Vecs | Sign Constraint | Test Error (%) |
|---|---|---|---|---|
| RNN | No | No | No | $1.65 \pm 0.27$ |
| RNN Vecs + ColEI Spectrum | Yes | No | No | $3.04 \pm 0.33$ |
| ColEI Vecs + RNN Spectrum | No | Yes | No | $2.32 \pm 0.35$ |
| RNN + Sign Constraint | No | No | Yes | $2.47 \pm 0.11$ |
| ColEI + No Sign Constraint | Yes | Yes | No | $3.61 \pm 0.47$ |
| RNN Vecs + ColEI Spectrum + Frozen Sign | Yes | No | Yes | $3.50 \pm 0.57$ |
| ColEI Vecs + RNN Spectrum + Frozen Sign | No | Yes | Yes | $2.72 \pm 0.27$ |
| ColEI | Yes | Yes | Yes | $3.95 \pm 0.45$ |

Table 1: Spectrum transplant experiment results for sequential MNIST. Table shows the final test error for all experiments in Section 2.3.1. Test error refers to the final sequential MNIST classification error. Results are averaged over 5 seeds.

| | Change Spectrum | Change Sing Vecs | Add Sign Constraint |
|---|---|---|---|
| RNN ablation **Error (%) Increase** | $+39.38 \pm 4.78$ | $-0.31 \pm 1.82$ | $+0.31 \pm 1.16$ |
| ColEI ablation **Error (%) Decrease** | $+44.06 \pm 5.71$ | $-1.56 \pm 7.84$ | $-7.81 \pm 12.58$ |

Table 2: Spectrum transplant experiment for the naturalistic object recognition task. Table shows the final test error. Note here changing singular vectors and sign constraint are not statistically different than having no impact (t-test, N=5). Results are averaged over 5 seeds.

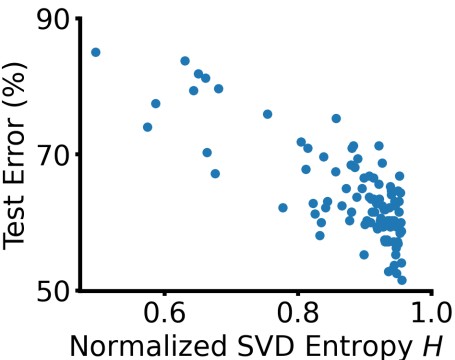

Figure 12: Final mean test error on the naturalistic object recognition task vs Normalised SVD entropy for all ColEI networks with the same configurations as Fig 7 (D) in the original paper.

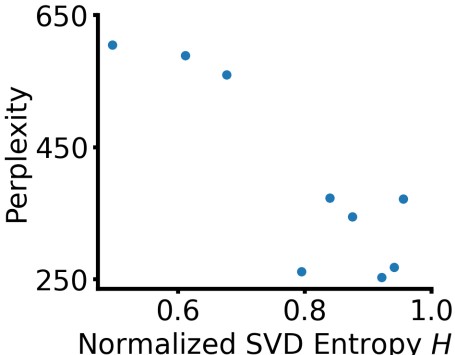

Figure 13: Final mean perplexity on the language modelling task (Penn Tree Bank) vs Normalised SVD entropy for 9 ColEI networks with size=100, 300, 1000; percentage of inhibitory cells=1%, 10%, 50%.

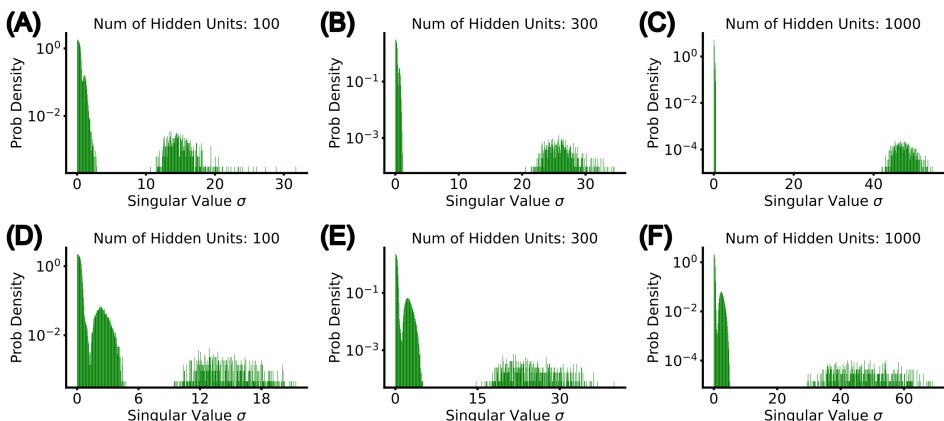

Figure 14: Histograms showing the distribution of singular values of the ColEI recurrent weights at initialization. (Top) Gamma distribution used to initialise ColEI. (Bottom) Uniform distribution used to initialise ColEI.

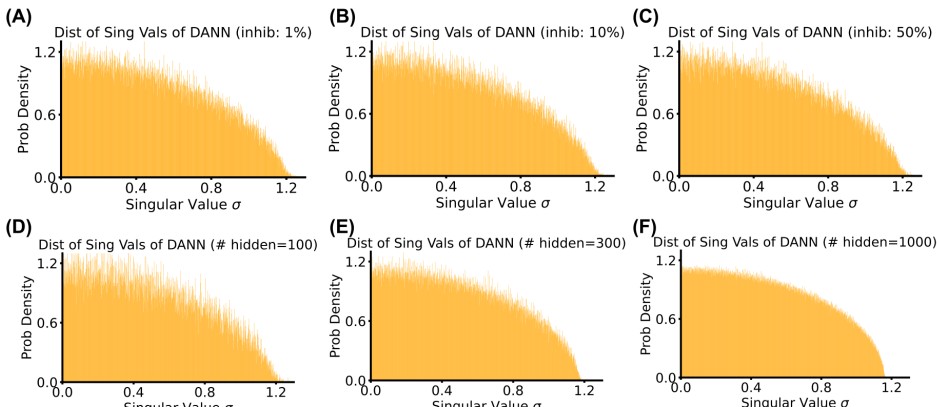

Figure 15: The DANN singular value spectrum is robust to the percentage of inhibitory cells and network size. Histograms showing the singular values of the DANN recurrent weight matrix at initialisation: (Top) Varying the ratio of E to I units. (Bottom) Varying network size.

|                 | $n = 100$ | $n = 300$ | $n = 1000$ |
|-----------------|-----------|-----------|------------|
| DANN 1% inhib   | 0.950     | 0.961     | 0.968      |
| DANN 10% inhib  | 0.950     | 0.961     | 0.968      |
| DANN 50% inhib  | 0.950     | 0.961     | 0.968      |
| RNN             | 0.952     | 0.962     | 0.969      |
| ColEI 1% inhib  | 0.413     | 0.604     | 0.679      |
| ColEI 10% inhib | 0.643     | 0.841     | 0.875      |
| ColEI 50% inhib | 0.921     | 0.941     | 0.956      |

Table 3: NSE values for networks with different sizes and EI ratios.

## 6.5 Detailed Discussion on the Origin of ColEI Multi-modal Spectrum

For intuition's sake, consider a $3 \times 3$ matrix $W$, with positive/negative columns, whose elements are sampled from a positive distribution $\delta$ with mean $\mu$ and variance $\sigma^2$, where the negative and positive items in a row are balanced (as is done in many ColEI models):

$$W \sim \begin{bmatrix} \delta(\mu, \sigma^2) & \delta(\mu, \sigma^2) & -\delta(2\mu, \sigma^2) \\ \delta(\mu, \sigma^2) & \delta(\mu, \sigma^2) & -\delta(2\mu, \sigma^2) \\ \delta(\mu, \sigma^2) & \delta(\mu, \sigma^2) & -\delta(2\mu, \sigma^2) \end{bmatrix}$$

Each element can then be rewritten as the sum of the means and a zero-mean random perturbation matrix, $P$, with variance $\sigma^2$:

$$W = \begin{bmatrix} \mu & \mu & -2\mu \\ \mu & \mu & -2\mu \\ \mu & \mu & -2\mu \end{bmatrix} + P$$

Although the distributions of singular values can't be simply added when two matrices are added, this analysis still provides some intuition behind the composite structure of ColEI's spectrum. We see that $W$ can be decomposed into (1) a rank one nilpotent matrix (since the square of it is a zero matrix) plus (2) a random zero-mean perturbation matrix. Notably, (1) has a large singular value ($\sigma_{max} = 3\sqrt{2}\mu$), yet all its eigenvalues are 0 (due to its nilpotency), whereas (2) contributes a unimodal distribution of singular values according to Marchenko-Pastur Law [22]. This example suggests that constraints on the eigenvalue spectral radius do not necessarily constrain the singular value distribution, and also provide clues on the multimodal structure of ColEI spectrum.

## 6.6 ColEI Initialisation Details

Following [10], the sign constraint at the input layer is realized at initialisation by independently sampling the input weights from an exponential distribution with variance $\frac{1}{d}$, i.e. $W_{in}^{(1)} \sim \exp(\sqrt{d})$ where $d$ is the dimension of the input. To centre the pre-activations the bias of the first recurrent layer $b^{(1)}$ is initialised as $b^{(1)} = -\sqrt{d}\bar{\mathcal{X}}$ where $\bar{\mathcal{X}}$ is the negative mean of the input dataset, pre-computed before training. Note that $\frac{1}{\sqrt{d}}$ is the expected value of the weights. For layers other than the input layer, the initial biases are all set to zero.

Next, the columns of all weight matrices other than the input ($W_{rec}^{\ell}$, $W_{in}^{\ell>1}$, and $W_{out}$,) are either all positive or all negative. These matrices are constructed at initialisation by first sampling a positive matrix which is then right multiplied by a non-random diagonal matrix $D$ consisting of entries of $1$ or $-1$, which specifies whether a neuron is excitatory or inhibitory. As with the input weight matrix the weights are sampled from exponential distributions. If multiple layers are stacked together, the column sign constraint always applies except for the input weights of the first layer since all its columns are positive. The initial hidden state $h_0 = 0$. However, in correspondence to regular initialisation schemes [25], we follow [10] and ensure that the means for the excitatory and inhibitory inputs to each neuron ($\mu_E$ and $\mu_I$) are balanced, and we also control the variance of the weights to not scale the variance of iid data as it passes through the network (see [10]). In other words, for any given neuron with inputs indexed by $j$, we have:

$$\sum_{j \in exc} |\mu_j| = \sum_{j \in inh} |\mu_j| \tag{6}$$

Therefore the exponential distribution used to initialise the excitatory weights has a variance of $1/(\frac{2\pi-1}{2\pi})(n_e + (\frac{n_e^2}{n_i}))$, where $n_e$ and $n_i$ represent the number of excitatory and inhibitory neurons in a layer. And the exponential distribution used to initialise inhibitory weights has a mean of $\mathbb{E}[w^+]\frac{n_e}{n_i}$, where $\mathbb{E}[w^+]$ is the mean of the excitatory distribution.

This above procedure generates $W_{in}^{\ell>1}, W_{out}$ and $W_{rec}^{\ell}$. Finally, we adjust recurrent weights to control their spectral radius, as is done in [9] and others. Specifically, we adjust the weights by setting:

$$W_{rec}^{\ell} \leftarrow (\rho/\rho_0)W_{rec}^{\ell} \tag{7}$$

where $\rho_0$ is the spectral radius of $W_{rec}^{\ell}$ before the adjustment. Note that the value $\rho$ determines the overall spectral radius, and inline with [9], we find that setting $\rho = 1.5$ often leads to the best performance for these networks. Since there are more excitatory neurons than inhibitory neurons (5-10 times) in cortical regions of the brain [24, 23], the default proportion of inhibitory neurons is 10% unless otherwise stated.