# OpenReview forum: "Learning better with Dale’s Law: A Spectral Perspective"
_NeurIPS.cc/2023/Conference — NeurIPS 2023 poster_

### Official Review · Reviewer_SnGV · 2023-07-03

**Soundness:** 3 good
**Presentation:** 3 good
**Contribution:** 2 fair
**Rating:** 6
**Confidence:** 4

**Summary:**

The author present a simulation study that investigates gradient-based optimisation of RNNs that obey Dale’s Law, i.e. networks with neurons that are strictly excitatory or inhibitory at initialisation and during training. In particular, the authors disentangle the effect of enforcing Dale’s Law during training (synaptic weights can not flip sign) and the initial spectral properties that result from initialisation with separate excitatory and inhibitory populations of neurons.

**Strengths:**

The paper investigates an important open question: What are the origins of the performance gap between RNNs that do / do not obey Dale’s Law. Through a large set of simulation studies and numerical analysis, the authors show in a convincing way that the performance gap does not solely stem from enforcing Dale’s Law during training (and thus restricting the solution space by enforcing signs of weights) but also significantly depends on how networks are initialised and parametrised. Initialising RNNs with columns of positive / negative weights to create populations of excitatory / inhibitory neurons (“ColEI networks”) causes a skewed and multi-modal singular value spectrum – leading to the well known effect of exploding neural activations / gradients that hamper training and performance. The authors further provide empirical evidence that the Normalised SVD Entropy thus provides an easy to compute predictor of network performance before the onset of training. Further, the authors provide analytical intuition for their simulation results in the discussion. The paper is well structured, written and accessible. The simulation details are explained in great detail; which is very helpful for understanding the research question, results and potential limitations. The figures are generally clear and accessible.

**Weaknesses:**

- My main concern is, that as written in appendix 5.3, the learning rate of DANNs is scaled independently for E and I weights to balance the impact and updates of E and I populations. As ColEI networks seem not to be trained with a similar scaling scheme, this has the potential to explain away or confound results and observations. I am happy to revise my score if this is adequately addressed.
- The authors base their analysis on ColEI networks that are initialised following [5], however, the authors do not provide insight (through simulations or analytically) if there generally can or cannot exist a weight initialisation scheme for ColEI networks which does not lead to a skewed and multimodal spectrum of singular values. As thus, the results seem to be limited to a specific initialisation scheme and not to ColEI networks in general.
- The authors apply gradient clipping during network training. This may significantly reduce the effects of skewed, multimodal singular value spectra and outliers in the singular value spectrum. As the authors try to disentangle the effects of enforcing Dale’s Law and the spectral properties of initial recurrent weight matrices, gradient clipping may confound results and conclusions.
- If I understand correctly, recurrent weight matrices in DANNs are parametrised by a linear combination of three weight matrices. As such, it seems like DANNs have three times more free parameters than ColEI networks. If true, this would pose the question if comparisons between DANNs and ColEIs as presented in figure 6 are fair.

Minor:
- In figure 2D it is difficult to see what is going on – maybe log-axis and a higher alpha value for the scatter point’s colour would help?
- In figure 2B it looks like the networks haven’t been trained until convergence
- Missing “in” in line 94
- Broken reference in line 110

**Questions:**

1) As written in appendix 5.3, the learning rate of DANNs are scaled independently for E and I weights to balance the impact of E and I and their updates. As ColEI networks seem not to be trained with a similar scaling scheme: How does the IE weight scaling influence simulation results and conclusions; in particular with respect to changes in EI ratio and network size?

2) In line 109 the authors write “[…] the activation variance did not scale with depth […]”. Did the authors control for mean and variance shift across layers and increasing depth? It would help to better understand the three different training regimes if there would be a plot that is visualising the mean and variance shift at initialisation across layers and depth for random data for RNNs, DANNs and ColEIs for varying EI ratios.

3) Related to 2 – In order to better understand the learning dynamics and in order to detect anomalies, a plot that visualises the mean and variance of synaptic weights and biases and how they evolve during training would be helpful. Do you expect the mean and variance of the weight matrices and biases within the E and I population to evolve similarly for RNNs, DANNs and ColEIs?

4) Related to 2 – Judging from the learning trajectories (e.g. in Figure 2B/C), to me, it looks like that ColEI networks have a significantly higher initial error (at t=0) – however, since the y-axis is cut – I might be wrong? If the initial errors are different, why are they different?

5) In line 116 the authors write “[…] each row of W^IE was initialised as the mean row of W^EE […]”. What does that mean? Do all entries in the row have the same value?

6) In Figure 3 it looks like ColEI networks have a large variance in performance. Do you have an explanation or intuition why some of the initialisations fail and others succeed?

7) From equation 3, I conclude that DANNs have 3x more free parameters in the recurrent part of the network than ColEIs. This makes comparison tricky, especially when it comes to network size. How does performance look like as a function of free parameters (instead of number of neurons in the hidden layer)?

8) I would assume that spectrum transplants alter the EI balance. Did you correct for that effect?

**Limitations:**

The authors have adequately addressed limitations of their work

---

> ### Author Rebuttal · Authors · 2023-08-09
>
> Many thanks to the reviewer for their thoughtful review. We are pleased they share our perspective that this paper investigates an important, unanswered question: the origins of the performance gap between EI and standard RNNs. Furthermore, it is our desire that this paper contributes to a broad spectrum of computational neuroscience research, so we are pleased that you find it well written, accessible and clear.
>
> Please see the general comment for the references list.
>
> **Weaknesses:**
> > My main concern is... …I am happy to revise my score if this is adequately addressed.
>
> In short, the scalings used for DANN models are specific to its architecture, not the parameters being E or I, and they do not apply to the ColEI network. In [1], the learning rates for DANN I parameters are scaled based on an analysis of how much an update to each parameter type (Wee, Wie, Wie) will change a layer’s output distribution, as quantified by KL divergence of the layer before and after the update.
>
> In a ColEI network, the E and I parameters have the same impact on the network’s output function (ColEI E and I parameters are like DANN Wee parameters), hence we did not originally run experiments with such scalings. However we agree this is valid scientific concern, and have ran additional ColEI experiments on sequential MNIST with outgoing (I->I, I->E) and incoming (E->I) I parameter-learning scaled as in DANNs (Rebuttal Fig 4).  We find that this scaling impairs learning, in line with our theoretical intuitions. We will add the figure to the appendix. We hope this addresses this important concern.
>
> >The authors base their analysis on ColEI...
>
> Thanks for this comment. To our knowledge, there is no better method for initializing ColEI networks [2, 3]. Nonetheless, we share the reviewer’s suspicion that an initialization exists for ColEI networks that results in better learning. As such, one contribution of our work is providing a potential foundation for this (currently) unknown method of initialization. Our results imply that if one can initialize ColEI networks with better spectral properties, then they should train well. We will add to the discussion highlighting this point.
>
> >The authors apply gradient clipping…
>
> Thank you for raising this. We missed an opportunity to highlight just how problematic the standard ColEI spectrum is! Gradient clipping is a common technique for training RNNs [4] and we applied it under the assumption that it would benefit ColEI networks. Indeed, we find that without gradient clipping ColEI networks perform very poorly, please see (Rebuttal Figure 3).
>
> >If I understand…
>
> DANNs do have more parameters than ColEI networks, but not as many as 3 times. Wei and Wie^T are both of dimension # hidden* # inhibitory. So for the case of 10% inhibitory units (Fig 6) there are only 20% more parameters. This is a relatively small difference, but if the reviewer wishes we can run and include experiments in which the total number of free parameters are equal. However in this case the number of hidden units and the dimensions of the hidden to hidden connectivity matrix would no longer be the same between DANNs and ColEI networks, and therefore these simulations would be unfair in a different way.
>
> Minor:
> >In figure 2D …
>
> We’ll add Fig 2d with zoomed in axes to the appendix (Rebuttal Fig 6A).
>
> >In figure 2B …
>
> Please see Rebuttal Fig 6B, we have increased the number of parameter updates from 5 to 18 M.
>
> **Questions**
> Thank you for this list, we will update the camera-ready version to address all of these points.
>
> >As written in appendix 5.3…
>
> Please see our comment in the weaknesses section. If desired we can include experiments with different ratios and network size, but we predict the ColEI performance will track the trends of the current results but be worse.
>
> >In line 109 the authors…
>
> Here, we just meant that we adopt the same general initialization strategy as found in [5,6].
>
> Regarding differences between hidden layer variance and mean over timesteps. We expect that there will indeed be differences, but that these will be another fingerprint of the spectral properties.
>
> >Related to 2 – In order…
>
> We expect there will be differences but expect it will be unclear how they relate to spectral properties. We are happy to discuss this and the previous point further if the reviewer wishes.
>
> >Related to 2 – Judging…
>
> All untrained networks have the same initial error, but the first point in the plot is the error after 100 updates. i.e. the error is not logged at the beginning.
>
> >In line 116 the authors…
>
> We apologize for an error in our text. We will change the text to “each row of W^EI was initialized as the mean row of W^EE”.  By this we mean that W^EI_ji = 1/n \sum_k W^EE_ki for all j, and I unit equivalence is then broken with W_ie weights. Therefore all entries in the columns of W^IE have the same values.
>
> >In Figure 3…
>
> Thank you for highlighting this aspect of ColEI training, we did reference it in a previous draft of the text and will reinsert this observation in the updated text. Our intuition is that variance in ColEI networks is due to them having fewer activity modes with appropriate singular values for learning computations over time (the motivation behind SVD entropy) and therefore small differences at initialisation can strongly impact learning.We also find that without gradient clipping ColEI networks are highly unstable, likely due to the distribution of singular values at initialisation [4], and this can also lead to variable performance given small differences at initialization.
>
>
> >From...
> Please see our response to this issue above.
>
> >I would…
> The spectrum transplant experiments convert EI RNNs into networks without separate populations of E and I units. We didn’t correct for this because to do so would require changing the spectral properties of the donor networks, and we would not be directly testing the impact of the different network spectral properties.

---

> > ### Comment · Reviewer_SnGV · 2023-08-15
> >
> > I would like to thank the authors for their thorough reply. Most of my concerns have been adequately addressed.
> >
> > Q2: While I agree that some of the differences in the mean and variance of activations would result from spectral properties, vanishing and exploding activations (and as a result, gradients) can also simply be the result of weights having a too small / large initial scale. Optimising the scale to ensure that the mean / variance stays as stable as possible at initialisation would be a simple way to dissociate the two.
> >
> > Q8: Did you consider evaluating how strongly the transplant experiments alter the EI balance, or at least to flag in the text that it does?

---

> > > ### Author Response · Authors · 2023-08-17
> > >
> > > We’re happy that our previous response managed to adequately address the majority of the reviewer’s concerns. Again, we would like to reiterate our thanks for their thorough and helpful review. We hope our answers below adequately address the reviewer’s final points.
> > >
> > > Finally, we encourage the reviewer to update their score in line with the degree of their increase in confidence.
> > >
> > > > I would like to thank the authors for their thorough reply. Most of my concerns have been adequately addressed.
> > > Q2: While I agree that some of the differences in the mean and variance of activations would result from spectral properties, vanishing and exploding activations (and as a result, gradients) can also simply be the result of weights having a too small / large initial scale. Optimising the scale to ensure that the mean / variance stays as stable as possible at initialisation would be a simple way to dissociate the two.
> > >
> > > Though we agree that regulating the mean and variance of activations is potentially useful in RNNs, the scale of the weights is part of what determines the spectral properties of the weight matrices, namely, the spectral radius, $\rho$ (Rajan & Abbott, 2006; Bordenave & Chafaï, 2012). Thus, in Figure 2D when we swept over different $\rho$ values, we were in fact sweeping over weight initialization scales. Note that ColEI networks are always worse than the best performing RNNs, no matter the scale of the weights. This strongly suggests that there is no scale of weights that would allow ColEI to perform as well as regular RNNs without fixing the other spectral properties. In fact, the $\rho$ = 1.5 scale used in Song et al. (2016), and throughout the rest of the figures, is selected in part to help keep the initial activations in an appropriate range and avoid vanishing and exploding activations/gradients as much as possible. We hope this clarifies the matter, but if we have misunderstood the reviewer’s point, please let us know.
> > >
> > > >Q8: Did you consider evaluating how strongly the transplant experiments alter the EI balance, or at least to flag in the text that it does?
> > >
> > > Thank you for re-raising this, we agree that it is valuable to evaluate and report any impact of these experiments on network EI balance. We therefore ran simulations and evaluated empirical balance at initialization (i.e. the mean of the sum of incoming weights to each neuron) and did not find a statistically significant difference. For a ColEI network with 10% inhibitory neurons and 1000 hidden units, the mean balance across units was -0.0006 with std. deviation 1.12, and for ColEI with the RNN spectrum it was 0.0696 with std. deviation 0.88 (t test, p=0.12). We will highlight this potential caveat and include these results in the text.

---

> > > > ### Comment · Reviewer_SnGV · 2023-08-17
> > > >
> > > > Thank you for (re-)addressing my questions. I have now revised my score.

---

### Official Review · Reviewer_gdj1 · 2023-07-06

**Soundness:** 3 good
**Presentation:** 3 good
**Contribution:** 3 good
**Rating:** 5
**Confidence:** 4

**Summary:**

The authors apply Dale's law (that neurons provide exclusively excitatory or inhibitory outputs) to RNNs with two different architectures: ColEIs which are the “straightforward” way of applying sign constraints per neuron, and recurrent DANNs based on an architecture with two layers per recurrent step. They compare singular value spectrums, network sizes and Excitation/Inhibition ratios in all three architectures and analyze how they affect performance. They conclude that the reason why DANNs perform as well as RNNs, while ColEIs perform worse is largely due to differences in the SVD distribution (as measured by Normalised SVD Entropy).

**Strengths:**

Dale’s law is an important and ubiquitous property of biological neural networks, but its consequences have not been thoroughly explored. It is interesting that a simple architecture can enforce Dale’s law while remaining trainable, and while keeping many properties of unconstrained RNNs. This architecture features feedforward inhibition, a biologically plausible motif. Understanding consequences of such biological structure is a key topic of Neuro-AI.

The work offers some nice empirical results, both in performance and in spectral properties.

The writing and figures are clear, and good efforts are made to provide explanations and interpretations.


**Weaknesses:**

While being advertised as a biological neural network, the architectural constraints that make the recurrent DANN trainable may be the very same ones that make it biologically unrealistic. For instance, there is no direct recurrent inhibition (I to I), and there cannot be direct reciprocal connectivity between E and I units.

There is not much mathematical analysis of the results, so the insights are somewhat limited. Clearly there are differences in the spectral distributions, but why these appear and most importantly how these are “directly responsible” for decreased performance is not deeply discussed. There is some discussion of the low-rank nilpotent component of ColEI matrices but there is more related literature to connect to, most notably from the Ostoijc group (no, this reviewer is not from that group). For example, Mastrogiuseppe and Ostojic 2018, Schuessler et al 2022, in addition to the Shao and Ostoijc 2022 paper the authors do cite (now evidently published in PLoS comp bio).

Many of the analyses are done only on the sequential MNIST task, which is not particularly natural. Claims are being made on the basis of small differences in errors, and some qualitative analysis. The small differences suggest that the networks are not being pushed very far to demonstrate their inductive biases.

Where did the ρ=1.5 and ρ ~= 1/sqrt(3) come from in Section 3.1?

How were the sign constraints implemented?

Minor: watch out for punctuation errors and typos.


**Questions:**

Are the inhibitory units in a DANN strictly linear? This seems necessary if the DANN weight matrix is actually an exact reparameterization of the RNN weight matrix, as advertised (L95). Either way, this is unclear. And this has important implications for the story: if the DANN is an exact reparameterization, then isn’t it trivial that RNN and DANN spectra are identical?
Why the variance of line 115? Why ρ=1.5 and ρ ~= 1/√3 in Section 3.1?
Can you explain the 3-layer architectures used in the Sequential MNIST figures? Do the eigenvalues and singular value visualizations of Figures 5,6 include values from weights from all layers? If so, are there any differences between the eigen/singular values between layers?
Line 200: Is initialization the only difference between a ColEI network without sign constraints and a standard RNNs?


**Limitations:**

Adequately addressed

---

> ### Author Rebuttal · Authors · 2023-08-09
>
> We thank the reviewer for their helpful review. Please see the general comment for the references list.
>
> > **Strengths**: Dale’s law is an important and ubiquitous property...
>
> Many thanks!
>
> **Weaknesses:**
>
> >While being advertised…
>
> Thank you for highlighting this aspect of DANNs, we will add to the text, noting that the lack of I-I connections in DANNs is not biologically realistic and that this could be changed in the future to make them more realistic, an important point of clarification. However, we want to highlight that the central motivation of this paper is to understand why EI networks can fail to train well, and our data supports the conclusion that the spectral properties of the weights is the main reason. Importantly, this central conclusion is not dependent on the specific DANN formulation being completely biologically faithful. The reviewer is correct that adding I-to-I connectivity would increase the biological faithfulness of recurrent DANNs. But the spectral properties would be the same, because fast-recurrent inhibition provides a “winner-take-all” mechanism, which would simply sparsify the activity. Therefore for simplicity we did not explore this aspect in our work.
>  But, the reviewer’s point holds, and we will be sure to highlight this biological infidelity of DANNs in our camera ready manuscript. We think it is an exciting direction of future research.
>
> One thing: we are unsure what the reviewer means by “there cannot be direct reciprocal connectivity between E and I units”, because the absence of I-I connections should not affect reciprocal E-I connectivity.
>
> >There is not much…
>
> We agree with the reviewer that our work is largely an empirical analysis rather than a mathematical analysis of the origins of poor learning in ColEI networks. We provide a discussion of the low-rank nilpotent matrices mainly for the purpose of intuitions. However, we would strongly contest that the empirical nature of our work means the insights are limited. As far as we know, the understandings we present here through our experiments are extremely novel and this is the first work that directly associates spectrum pathology with poor ColEI learning performance. Overall, this work is a major shift in the field’s understanding of why recurrent networks of E and I units perform poorly.
>
> We also wish to distinguish our study from prior works like Mastrogiuseppe and Ostojic 2018, and Schuessler et al 2022. First, our work focused on RNNs respecting Dale’s law whereas the mentioned works did not. Also, the mentioned works mainly focused on exploring the relationship between low-rank connectivity, (low-rank) network dynamics and how network computations are related to them, rather than how to understand the learning performance of E-I RNNs. In contrast, in this work we aim to directly link singular value spectrum with the learning performance of E-I RNNs when we train them with gradient descent. Past research also mainly looks at the spectral properties of random networks, with little focus on learning performance [7,8] Therefore, we hope this work can lay the foundation for future research on optimizing network performance without breaking biological constraints. But we agree the mentioned works are relevant and we are happy to cite them.
>
> >Many of the analyses …
>
> Thank you for highlighting this. We agree it is very important to verify that our results and analyses are consistent across data distributions, especially more naturalistic ones given this work’s connections to neurobiology. As a result of your comment we have included a new naturalistic image task and also ran the analyses presented in Figures 5 and 6 on the Penn Treebank. Please see the general response for more details. We believe these experiments substantially improve the paper - thank you again for suggesting this.
>
> >Where did the ρ=1.5 and ρ ~= 1/sqrt(3) come from in Section 3.1?
>
> Rho = 1.5 is from [2]. 1/root(3) comes from  Pytorch’s default initialization for RNNs.
> How were the sign constraints implemented?
> Sign constraints are implemented via projected gradient descent. After a parameter update outgoing parameters from E or I neurons are clamped to either be all positive or all negative respectively.
>
> **Questions:**
> >Are the inhibitory units…
>
> Thank you for raising this. As noted in [1], the nonlinearity used for inhibitory units in DANN is ReLU. The subtlety here is that the post-activation from the previous layer of DANN is non-negative due to ReLU,  rendering the pre-activation of inhibitory units of DANN non-negative as well (because Wei is a non-negative matrix). Therefore, ReLU and linear activation functions are equivalent for the inhibitory units in DANN. The expression in line 95 is a simplification of the subtlety explained above. In addition, we wish to emphasize that the equivalence of RNN and DANN spectra is nontrivial because both the low-rank matrix Wei*Wie and the fully positive matrix Wee in DANNs can lead to outliers in the spectrum at initialization [9] and must be carefully chosen in order to cancel out each other. Rebuttal Fig. 5 shows the spectrum of a poorly initialized DANN weight matrix although all elements of the net-effect W=Wee - Wei*Wie matrix have mean zero.
>
> >Why the variance of line 115? Why ρ=1.5 and ρ ~= 1/√3 in Section 3.1?
>
> Please see above and also footnote in the main text (page 5) for the correspondence between variance and rho.
>
> >Can you explain the 3-layer architectures…
>
> By three layers we mean that there are 3 RNN modules stacked on top of each other,
>
> >Do the eigenvalues …between layers?
>
> All layers have the same number of recurrent units and are initialized with the same random initialization scheme, therefore they are the same.
>
> >Line 200: Is initialization …RNNs?
>
> Yes, correct!

---

### Official Review · Reviewer_YJwG · 2023-07-07

**Soundness:** 3 good
**Presentation:** 4 excellent
**Contribution:** 3 good
**Rating:** 6
**Confidence:** 4

**Summary:**

The paper investigated the problem of why columnEI networks have impaired learning, and they experimentally found that instead of sign constraint, the spectral property of weight at initialization contributed the most; they further experimentally showed that E/I ratio and network size change the spectral properties thus lead to different learning performances. Additionally, they showed DANN in the RNN form show similar performance as normal RNN in 3 different tasks.

**Strengths:**

**Originality** This is a follow-up work on applying DANN to RNN. I appreciate the detailed discussion on why it is hard to train E-I separated RNNs.

**Quality**The effectiveness of DRNN is tested in three different tasks which is great! And all hyperparameter selections are listed in the appe ndix. The discussion on initialization spectral property contain proper ablation experiments and extended discussion on E-I ratio and network size.

**Clarity** The paper is clearly written, with details in the appendix. Some part of the result explanation can be confusing (see second question)

**Significance** I believe the paper will be of interest to the computational neuroscience community.

**Weaknesses:**

- The discussion of initialization spectral property is only done for sequential MNIST. Does the observation on SVD entropy hold across data distributions? I'm a bit concerned on how generalizable the results are (details see question).
- For the sign constrained training, how are the gradients rectified? Set to zero? If set to zero, it may leads to silent unit problem. Did the authors exclude the possibility that sign constrained RNN learn worse due to increased number of silent units?
- Typo: line 110 citation

**Questions:**

My main concern is on the spectral discussion
- Does it extend to other tasks with different data distribution?
- Why clusterness of singular values matter for learning? The intuition currently given in the paper seems to mainly pertain to large singular values; yet it is shown large singular value ColEI can learn well. Then what is the intuition behind SVD entropy tracking performance? This lack of intuition is what prompted me to ask the first question as it is not immediately clear to me how generalizable the current results are. And thus a current score of 5.

**Limitations:**

See questions

---

> ### Author Rebuttal · Authors · 2023-08-09
>
> We thank the reviewer for their helpful review. Please see the general comment for the references list.
>
> >**Weaknesses** The discussion of initialization spectral property is only done for sequential MNIST. Does the observation on SVD entropy hold across data distributions? I'm a bit concerned on how generalizable the results are (details see question).
>
> Thank you for raising this. We agree it is very important to verify that SVD entropy is predictive of performance across different data distributions. We have therefore run more experiments (Penn Treebank and the new naturalistic image task) to verify the generalizability of our results. Please see the global response and the Rebuttal Figures. Thank you for this comment, it has helped us improve the paper.
>
> >For the sign constrained training, how are the gradients rectified? Set to zero? If set to zero, it may leads to silent unit problem. Did the authors exclude the possibility that sign constrained RNN learn worse due to increased number of silent units?
>
> We would like to clarify that we do not rectify the gradients. Sign constraints are realized using projected gradient descent, i.e. after a parameter update via gradient descent as normal, parameters were then clamped to be all positive or all negative depending on whether the parameters were E or I. We will make this clear in the methods section.
>
> In response to the reviewer’s second comment we investigated the distribution of weights and found that the sparsity of the weight matrices for ColEI networks and regular RNNs are very similar, making silent units from sign constraints unlikely to be an issue. This observation also aligns with our observation that sign constraints have a minimal effect on learning performance.
>
> > **Questions:** My main concern is on the spectral discussion.  Does it extend to other tasks with different data distribution?
>
> Please see our response to this question above.
>
> > Why clusterness of singular values matter for learning? The intuition currently given in the paper seems to mainly pertain to large singular values; yet it is shown large singular value ColEI can learn well. Then what is the intuition behind SVD entropy tracking performance? This lack of intuition is what prompted me to ask the first question as it is not immediately clear to me how generalizable the current results are. And thus a current score of 5.
>
> Thank you for communicating this point. We originally shared the reviewer’s puzzlement, as larger ColEI networks have even larger maximum singular values (sigma_max) and yet train better.  Indeed this observation led us to propose normalized SVD entropy as a candidate metric for tracking performance and spectrum pathology. As explained in the main text (lines 279 - 287), the intuition behind SVD entropy tracking performance is that the bulk of the singular values also matters [10,11]. Put another way, for the large networks, although there are a small number of directions with a large singular value, the bulk of them are smaller, which helps ameliorate the problem of exploding gradients, by rotating the activity away from the direction corresponding to the large singular values. As such, to understand learning performance we require a metric that captures the overall effect of the entire singular value spectrum, rather than just a few singular values, which we do using the SVD entropy metric.
>
> Thanks to the reviewer’s comment, we realize that the current manuscript may benefit from describing this insight earlier in the results, rather than in the discussion (lines 279 - 287) after the results are presented. We will also note that the additional experiments with the Penn Treebank and the naturalistic images further demonstrate the validity of using SVD entropy to track ColEI performance.

---

> > ### Comment · Reviewer_YJwG · 2023-08-18
> >
> > I would like to thank the authors for their response and added experiments! My major concern with generalizability is now resolved.  With the added experimental results, I believe the paper is now stronger thus increased my score accordingly.
> >
> > > Intuition explanation
> >
> > The rebuttal made the author's point very clear! I'd suggest substitute part of lines 279 - 287 with the writing in the rebuttal.
> >
> > > Sign constraint implementation
> >
> > I understand they are clamped, but clamped to what value specifically? Say SGD updates w_ij from 0.1 to -0.1, then is it clamped to 0 or 0.1 or some other positive value? If it's clamped to 0, then the problem of silent unit stands. Also, to check for silent unit problem, I'd check for activity sparsity instead of weight sparsity. I would encourage the authors to make my first part of the question clear in the final version.

---

> > > ### Author Response · Authors · 2023-08-18
> > >
> > > We would like to express our gratitude again for your insightful feedback. We will change the main text as suggested and hope our responses below could adequately address the reviewer's remaining concerns.
> > > - To clarify sign constraint implementation: Yes, if $w_{ij}$ is initialized as 0.1 and SDG attempts to change it to -0.1, then we clamp it at 0. But if the update given by another mini-batch attempts to change $w_{ij}$ back to positive weights, e.g. 0.05, then $w_{ij}$ will change to 0.05 as normal.
> > > - To exclude the possibility of a silent unit problem due to sign constraint, we ran experiments on the sequential MNIST task using ColEI networks with/without sign constraints, and we checked if there was any silent units every 100 updates during training. Interestingly, we did not find any silent units in all checkpoints regardless of sign constraints. Our intuition is that to be a silent unit in an RNN, a neuron must be inactive for all datapoints and across all timesteps, yet in feedforward networks silent units only need to be inactive for all datapoints. Therefore, intuitively it should be much harder to have silent units in RNNs than in feedforward networks.

---

> > > > ### Comment · Reviewer_YJwG · 2023-08-19
> > > >
> > > > Thank you for your response and explanations! All of my concerns are resolved now.

---

### Official Review · Reviewer_TD5G · 2023-07-21

**Soundness:** 3 good
**Presentation:** 3 good
**Contribution:** 2 fair
**Rating:** 5
**Confidence:** 4

**Summary:**

This paper presents a comparison between the performance of standard RNNs and two classes of models: the ColEI network and the DANN, which incorporate Dale's Law with the constraint that units in a circuit should be excitatory or inhibitory but not both. The DANN achieves similar performance to a non-signed RNN, but ColEI shows inferior performance. The paper demonstrates that the spectral properties of the recurrent weight matrix at initialization have a more significant impact on network performance than sign constraints, which may explain why some forms of EI network learn better than others.

**Strengths:**

- This paper uses analytical approaches from machine learning to explore network models that adhere to biological principles, which may inspire the development of more biologically realistic models that also have good performance.
- A novel contribution is the introduction of normalised SVD entropy as a measurement of spectrum pathology during the initialization stage which predicts the final performance of a network before training.
- It also includes extensive experiments that test three RNN architecture on three different tasks with progressive difficulty levels.

**Weaknesses:**

This work builds upon existing research about EI networks with incremental advancements, offering interesting insights into how the spectral properties of the recurrent weight matrix at initialization may impact network performance. However, the practical application for designing EI networks that perform well requires further clarification. It is unclear to me whether DANNs can be effectively improved with this insight, and even with a more appropriate spectrum, ColEI falls short of standard RNN or DANN performance.

**Questions:**

Considering the biological motivation of having dedicated excitatory and inhibitory units, it would be valuable to provide a deeper validation or interpretation of the results in the context of neurobiology. For example, could the finding about the different effects of changing the ratio of E/I units on ColEI networks and DANNs offer insights into the biological relevance of these two models to the neural circuit in the brain?

**Limitations:**

While the focus on RNNs is relevant to the paper's objectives, it would be helpful to briefly acknowledge potential implications for other types of networks.

---

> ### Author Rebuttal · Authors · 2023-08-09
>
> We thank the reviewer for their review. Please note that the reference numbers refer to the references list in the general comment.
>
> >**Weaknesses:** However, the practical application for designing EI networks that perform well requires further clarification. It is unclear to me whether DANNs can be effectively improved with this insight, and even with a more appropriate spectrum, ColEI falls short of standard RNN or DANN performance.
>
> Thank you for this comment, we agree that adding a paragraph to the discussion that clarifies and expands on the practical applications of our results to future work would improve the paper.
>
> First, the novel insight that the spectral properties are the main issue for EI network learning vs standard networks can indeed be applied to DANNs. We can improve the DANN spectrum by carefully selecting Wee, Wei, and Wie to approach an orthogonal net-effect matrix W=Wee - Wei Wie with singular values~=1, which has been shown to be helpful for RNNs [10,11]. We have data showing improved performance on sequential MNIST from this strategy. We did not include this direction in the original draft due to the page limitation and our focus on explaining poor training in ColEI networks, but we can add it to the appendix if helpful.
>
> Second, it is true that ColEI networks fall short of standard RNN and DANN performance, even with hyperparameter choices that make the spectrum more appropriate (i.e. larger sizes and ratios of E to I). However, the insight that it is the spectrum that underlies the poor ColEI performance is a foundational one. We hope our work serves as the basis and inspiration for an alternative initialization for ColEI networks that improves their learning. We will add a comment in the discussion highlighting this as a future research direction.
>
> Finally, another exciting future direction is to hybridize DANNs and ColEI networks, as DANNs will remove the far-outlying singular values of the ColEI spectrum and will have the effect of enabling sign changes in the effective recurrent weight matrix (which still have a minor contribution to poor ColEI performance). This hybridization would also be satisfying from a neuroscientific standpoint as the inhibitory neurons in ColEI networks can be thought of as modeling different inhibitory populations (eg. SOM+, CCK+ etc. cells) than the fast inhibitory populations modeled by DANNs (PV+ cells). Again, we omitted this discussion originally due to the page limit but believe this is a very promising direction and would be happy to add some discussion of it.
>
> > **Questions:** Considering the biological motivation of having dedicated excitatory and inhibitory units, it would be valuable to provide a deeper validation or interpretation of the results in the context of neurobiology. For example, could the finding about the different effects of changing the ratio of E/I units on ColEI networks and DANNs offer insights into the biological relevance of these two models to the neural circuit in the brain?
>
> We agree, and will strengthen the neurobiological focus for the camera ready version of the paper. Related to our response above, we think that the two models have slightly different neurobiological interpretations as the populations of inhibitory neurons that they naturally model are different: ColEI RNNs are a more general model of I cells, whereas DANNs specifically model fast PV+ interneurons. Interestingly, and to the reviewer’s point, our finding that ColEI networks learn better with more inhibitory cells (a smaller E/I ratio), and that DANNs learn fine even with very large E/I ratios, fits with the small number of PV+ cells compared to the total pool of inhibitory cells (~20%) [12] in real neural circuits. Furthermore, recent work has found that the human and macaque cortex contains approximately three times the amount of inhibitory interneurons than mice [13], which is consistent with our finding that a more balanced ratio of E to I results in better spectrum and learning for ColEI networks.
>
> >**Limitations:**
> While the focus on RNNs is relevant to the paper's objectives, it would be helpful to briefly acknowledge potential implications for other types of networks.
>
> Thank you for pointing this out. We will edit the text highlighting how we expect these insights to apply to all EI networks: e.g. feedforward MLPs, convolutional networks and potentially spiking neural networks.

---

### Author Rebuttal · Authors · 2023-08-09

We thank all reviewers for their time and effort in providing a set of positive, high-quality helpful reviews.

We would like to take this opportunity to reiterate the main contribution of our work. We investigate the following question: why do recurrent neural networks (RNNs) that have separate excitatory (E) and inhibitory (I) neurons often fail to train as well as regular RNNs? We present evidence that strongly supports the theory that, instead of sign constraints, the traditional column E-I network design is handicapped because the synaptic weight spectrum is multimodal and disperse, unlike standard RNNs.  We also show that there are E-I networks based on fast inhibition (Dale’s ANNs or DANNs) that have spectral properties akin to standard RNNs, and these networks train well. This work substantially shifts our understanding of the origins of poor learning in column EI networks, and suggests that contrary to popular assumptions, the imposition of sign constraints on a neural network is not the main impediment to good training.

Here we communicate the more general and most substantial improvements we have made as a result of reviewer feedback:

* We have added a new naturalistic image categorization task and verified that all our results and analyses apply in this new setting (see “Natural object recognition task description” below and Rebuttal Fig 1). Consistent with our previous findings, spectrum transplant experiments decrease/increase error by ~40% whereas adding/removing sign constraints only account for ~0.3 and 8% error differences respectively (Rebuttal Fig 1C & Table 1).
* Similarly, we have extended the sequential MNIST results presented in Fig 5, 6 to the Penn Treebank dataset which is both more challenging and more naturalistic (as it is a natural language processing dataset). Again, results here are similar to sequential MNIST; we find that normalized SVD entropy is anti-correlated with perplexity.
* We have run additional experiments without gradient clipping, which showcase more obviously the problematic spectral properties of ColEI networks.

We believe that these additions attend to some of the most pressing questions raised by the reviewers and strengthen the conclusions of the paper. We also hope each of the individual, more directed responses adequately address the additional reviewer comments, and we look forward to the discussion in the next phase. Due to the new character limit per response we have had to prioritize brevity for some of the discussion points, so please do not hesitate to request further clarification.

## Natural object recognition task description:

Following Majaj & Hong, et al., 2015, we designed the following naturalistic object recognition task for RNN: at the first time step, the input to the RNN is a flattened achromatic image of one of 64 natural objects from 8 categories (e.g., Fruits, Faces) with a random natural scene background. Then the network receives no inputs and must retain information about the image over 5 timesteps to output the classification result at the last time step. The task presents two main challenges: retaining the information over time without additional inputs and correctly mapping different natural objects, like raspberry and watermelon, to the same category, such as Fruits. Please see
Majaj & Hong, 2015 for more details. https://www.jneurosci.org/content/jneuro/35/39/13402.full.pdf

## References:

[1] Cornford et al 2021
[2] Song et al, 2016
[3] Yang & Wang, 2021
[4] Pascanu et al., 2012
[5] He et al., 2015
[6] Glorot & Bengio, 2010
[7] Rajan & Abbott, 2006
[8] Harris et al., 2023
[9] Benaych-Georges & Nadakuditi 2012
[10] Arjovsky et al., 2016
[11] Le et al., 2015
[12] Bezaire & Soltesz, 2016
[13] Loomba et al., 2022

---

### Decision · Program_Chairs · 2023-09-21

**Decision:**

Accept (poster)

**Comment:**

The study investigates why training recurrent neural networks (RNNs) with excitatory (E) and inhibitory (I) neurons is harder compared to regular RNNs. Authors demonstrate that the problem is not sign constraints, but rather the multimodal and disperse synaptic weight spectrum in the traditional E-I network design. Reviewers and the AC found this result to be important. Most criticism was addressed during the rebuttal period and additional experiments strengthened the paper.